# ReCAP: Recursive Prompting for Self-Supervised Category-Level Articulated Pose Estimation from an Image

## Abstract

Estimating category-level articulated object poses is crucial for robotics and virtual reality. Prior works either rely on costly annotations, limiting scalability, or depend on auxiliary signals such as dense RGB-D sensing and geometric constraints that are rarely available in practice. As a result, articulated pose estimation from a single RGB image remains largely unsolved. We propose **ReCAP**, a **Re**cursive prompting for self-supervised **C**ategory-level **A**rticulated object **P**ose estimation from an image. ReCAP adapts a pre-trained foundation model using a Recursive Prompt Generator with residual injection, introducing less than 1% additional parameters. This mechanism enables parameter-efficient scaling through recursive refinement, while residual injection preserves token alignment under dynamic reconfiguration, yielding robust articulated-object adaptation. To further resolve structural ambiguities, we introduce $\mathcal{X}$-SGP, a multi-scale fusion module that adaptively integrates semantic and geometric cues, an aspect often overlooked by geometry-centric approaches. Experiments on synthetic and real benchmarks demonstrate state-of-the-art monocular articulated pose estimation without requiring 3D supervision or auxiliary depth input. To the best of our knowledge, ReCAP is the first self-supervised framework to accomplish this task from a single image.

## 1 Introduction

Accurate estimation of articulated objects from visual signals is crucial for both intelligent robots and next-generation AR experiences (Chang et al., 2020). Due to geometric uncertainty arising from depth ambiguity and occlusion, early works often rely on high-quality annotation (Li et al., 2020) or incorporate expanded signals such as dense RGB-D sensing or multi-view 3D supervision (Li et al., 2020; Yu et al., 2024a; Fu et al., 2024). While these methods have advanced the field, they remain constrained by three major limitations (**LI**–**LIII**), which we elaborate on below.

Research on category-level articulated object pose estimation spans supervised, weakly supervised, and self-supervised paradigms. Supervised (Li et al., 2020; Jiang et al., 2022; Yu et al., 2024b) and weakly supervised (Zheng et al., 2023) methods suffer from (**LI**) *prohibitive annotation costs and poor scalability*, as they require large-scale, high-quality joint-level labels that are difficult to obtain given the complexity and diversity of articulated pose configurations. Self-supervised methods (Liu et al., 2023b; Che et al., 2024) alleviate labeling demands but incur (**LII**) *unrealistic data assumptions*, typically assume readily available depth point clouds as input (Xue et al., 2021; Xiang et al., 2020), which are rarely accessible outside synthetic environments. Fundamentally, most methods depend solely on geometric cues, leading to (**LIII**) *limited semantic reasoning*, resulting in structural ambiguities and unreliable joint correspondences. As a result, models remain vulnerable to occlusion and sensor noise, even when equipped with proxy losses (e.g., Chamfer Distance) within SE(3)-equivariant frameworks (Li et al., 2021).

Given these limitations, inferring articulated object poses directly from a single in-the-wild RGB image remains a fundamental yet unsolved challenge. Without depth, multi-view supervision, or curated annotations, models face severe depth ambiguity, occlusion, and diverse visual appearances, under which geometry-centric paradigms inevitably break down.

In this paper, we present **ReCAP**, the first self-supervised framework for single-image category-level articulated object pose estimation, designed to simultaneously address challenges **(LI)**–**(LIII)**. Although the recent geometry foundation model VGGT (Wang et al., 2025) offers strong monocular 3D priors, making it a promising starting point for articulated pose estimation, its rigid-object pre-training (Reizenstein et al., 2021; Ling et al., 2024) limits part semantics and kinematic variability, causing distribution shifts on articulated targets. Adapting the foundation model (12.6B parameters) by naive finetuning is costly and risks harming generalization (Wortsman et al., 2022). We therefore adopt embedding-space adaptation via vision prompting (Raghu et al., 2021; Yang et al., 2022). However, applying prepended-token prompting (Jia et al., 2022) to VGGT exposes two limitations in stability and scalability. Token reconfiguration prevents prepended tokens from providing consistent benefits, while shallow prompts lack expressiveness and deeper stacking inflates parameters, worsening the capacity–efficiency trade-off. To overcome these issues, we propose the Recursive Residual Prompting. Specifically, recursive refinement enables parameter-efficient scaling, while residual injection stabilizes prompts under misalignment, ensuring robust adaptation.

After adapting to articulated objects with prompts, we further address occlusion and structural variation, factors often overlooked by geometry-centric approaches (Che et al., 2024; Li et al., 2021), by developing the Cross Semantic–Geometry Pyramid ($\mathcal{X}$-SGP), which hierarchically fuses semantic and geometric cues via adaptive modulation. To additionally ensure robustness across viewpoints, we enforce a geometry-invariant feature space through discrete alignment and normalization. Finally, we employ a unified PoseHead (Che et al., 2024) to jointly estimate object- and part-level poses, transforming rigid-object priors into a semantics- and geometry-aware representation for robust self-supervised monocular articulated pose estimation.

Our contributions can be summarized as follows:

- We introduce the Recursive Residual Prompting that overcomes the limitations of standard prompt tuning under dynamic token reconfiguration. It also achieves parameter-efficient, depth-equivalent adaptation of VGGT to articulated objects.

- We introduce a hierarchical semantic–geometry fusion module that adaptively resolves occlusion and symmetry ambiguities overlooked by geometry-centric methods.

- To the best of our knowledge, ReCAP is the first self-supervised framework for category-level articulated object pose estimation from a single RGB image. It achieves state-of-the-art results on both synthetic and real-world articulated object benchmarks, paving the way for practical monocular articulated object pose estimation.

## 2 RELATED WORK

**Category-level Articulated Object Pose Estimation.** Articulated object pose estimation targets part-level pose, segmentation, and joint recovery for movable category-level objects. Early supervised methods (Li et al., 2020; Jiang et al., 2022; Yu et al., 2024b; Zhang et al., 2024a) achieved impressive results using large-scale datasets (Xiang et al., 2020; Liu et al., 2022b; Xue et al., 2021; Liu et al., 2022a) with dense part and joint annotations. In parallel, render-and-compare approaches (Liu et al., 2022a; Zhang et al., 2025) estimate poses by minimizing the discrepancy between rendered and observed views. However, such annotation requirements and complex data preparation hinder scalability. To alleviate annotation demands, recent works explored self- and weakly supervised approaches. EAP (Liu et al., 2023b) enables self-supervised part segmentation and pose estimation from single-frame point clouds, but struggles with large geometric and pose variations, often leading to inconsistent segmentation. OP-Align (Che et al., 2024) improves robustness to pose variance via part-level alignment, but relies on high-quality point clouds and uniform object structures, limiting applicability in real-world scenarios. Other lines exploit multi-view observations (Insafutdinov & Dosovitskiy, 2018; Li et al., 2018) or neural implicit representations (Irshad et al., 2022; Zhang et al., 2024b) to recover articulation without direct pose supervision. Despite these advances, most methods still rely on depth sensors and emphasize geometry while neglecting semantic cues. Although some works explore single-image articulated object generation or reconstruction (Lu et al., 2025; Aygun & Mac Aodha, 2024), they do not tackle category-level pose estimation from raw RGB. This gap motivates investigating articulated pose estimation directly from a single RGB image by leveraging semantic and geometric cues.

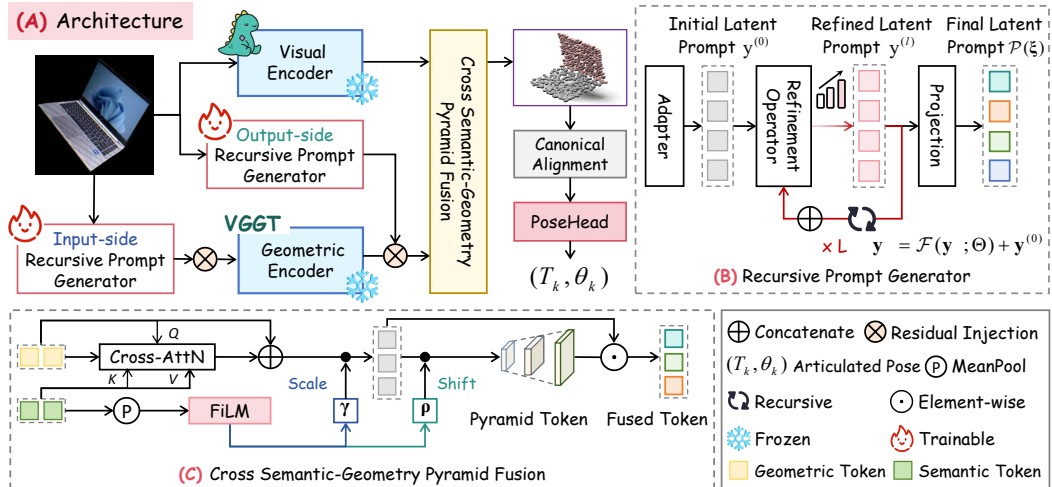

Figure 1: Overview. (A) *System Pipeline*: Given a single image $\mathbf{I}$, the network predicts articulated poses $(T_k, \theta_k)$. It integrates a Visual Encoder (DINOv2) and a Geometric Encoder (VGGT), the latter guided by Recursive Prompt Generators (input- and output-side, Sec. 3.2). Features are fused by the Cross Semantic–Geometry Pyramid Fusion module (Sec. 3.3), then passed through Canonical Alignment and the final pose estimation head (Sec. 3.4). (B) *Recursive Prompt Generator*: Generates prompts through iterative refinement, starting from an initial latent prompt, the Refinement Operator recursively updates the representation, and a Projection produces the final latent prompt. (C) *Cross Semantic–Geometry Pyramid Fusion*: Combines semantic and geometric tokens via cross-attention and FiLM modulation, applying scale and shift operations to produce fused tokens.

**Prompt Learning in Computer Vision.** Prompt learning originated in NLP to adapt large language models with most parameters frozen. As part of the broader family of parameter-efficient fine-tuning (PEFT) methods (Liu et al., 2023a; 2021; 2024), it has been extended to vision-language models such as CLIP (Radford et al., 2021) and further developed into continuous and conditional prompts (Zhou et al., 2022b;a). In vision, prompt tuning has emerged as an efficient alternative to full fine-tuning for transformers (Jia et al., 2022; Park & Byun, 2024), and has been explored for point cloud understanding (Zhou et al., 2024) and multi-modal learning (Yang et al., 2022; Wang et al., 2024a). Despite progress on prompt-based human pose (Li et al., 2025) and hand pose (Garg et al., 2024), articulated object pose estimation from a single RGB image under self-supervision remains largely unexplored, which demands structured 3D reasoning and articulation-aware adaptation.

## 3 METHOD

### 3.1 TASK FORMULATION

We tackle category-level articulated pose estimation from a single RGB image $\mathbf{I}$ in a self-supervised setup without depth, multi-view, or 3D annotations. We assume the part count $K$ and joint types (revolute or prismatic) are known per category, but no instance-level labels (e.g., part masks, correspondences, poses) are given. For each of the $K$ parts, we predict its 6-DoF rigid transformation:

$$T_k = [R_k \mid t_k] \in SE(3), \quad R_k \in SO(3), \; t_k \in \mathbb{R}^3.$$

together with its articulation parameter $\theta_k$ (e.g., revolute angle or prismatic displacement). Formally, the task is to learn a mapping:

$$f_\phi : \mathbf{I} \mapsto \{(T_k, \theta_k)\}_{k=1}^K,$$

where articulated pose is defined by the set of rigid part transformations and their articulation states.

### 3.2 RECURSIVE RESIDUAL PROMPTING

**Observation and Motivation.** We tackle category-level articulated pose estimation in the challenging self-supervised, single-RGB setting. To obtain geometric priors without depth supervision, we

build on the geometry foundation model VGGT (Wang et al., 2025), which predicts dense point clouds from monocular RGB. However, since VGGT is mainly trained on rigid objects (Reizenstein et al., 2021; Ling et al., 2024), it fails to capture part semantics and kinematic variability, resulting in a severe distribution shift on articulated targets (see Table 4). Direct fine-tuning of its 12.6B parameters is computationally prohibitive and risks catastrophic forgetting (Dong et al., 2024), motivating a prompt-based adaptation strategy.

Basically, prompt tuning (Li & Liang, 2021; Yang et al., 2022; Jia et al., 2022) adapts pretrained models by prepending learnable prompts to task tokens. However, two issues prevent its direct application to VGGT in our task: **1) *Limited expressivity of shallow prompts*.** Shallow prompts have limited capacity to model complex representations. Increasing their number or stacking them across layers can improve expressivity, but incurs significant fine-tuning cost, undermining the efficiency advantage of prompt tuning. **2) *Dynamic token misalignment*.** Prompt tuning assumes that tokens remain statically aligned across layers, preserving a fixed correspondence as information propagates through the network:

$$\mathcal{T}_i = [\mathbf{E}_i;\ \mathbf{P}_i], \quad \mathcal{T}_i \text{ maintains index alignment with } \mathcal{T}_{i-1}, \tag{1}$$

where $\mathbf{E}_i$ (task) and $\mathbf{P}_i$ (prompt) are aligned to preserve cross-layer correspondence. This assumption is violated in architectures with dynamic token operations (Yang et al., 2023; Li et al., 2024), including VGGT, where token sets are reconfigured during propagation. As a result, prompts lose consistent correspondence with task tokens, limiting effectiveness, as also observed in (Jia et al., 2022). We substantiate this theoretically in Appendix B.1 and empirically in Appendix B.2.

**Proposed Recursive Residual Prompting.** To address the above two challenges, we introduce Recursive Residual Prompting, which tackles the limited capacity of shallow prompts through ***recursive update*** and injects prompts to the backbone's stable boundary layers via ***residual injection*** to mitigate dynamic token misalignment. Finally, the prompts are injected at both the input and output sides of the backbone with our ***complementary dual prompts*** design. We present the details of the three key modules as follows.

**(i) *Recursive update*.** As shown in Fig. 1(B), the structure of the prompt generator is defined via truncated fixed-point iteration, with $\boldsymbol{\xi} \in \mathbb{R}^{N \times d}$ as anchor features, $\mathbf{y}^{(l)} \in \mathbb{R}^{N \times d}$ the refined latent prompt, and shared parameters $\Theta$ across iterations:

$$
\begin{aligned}
\mathbf{y}^{(0)} &= \text{Adapter}(\boldsymbol{\xi}), \\
\mathbf{y}^{(l)} &= \mathcal{F}\Big(\mathbf{y}^{(l-1)}; \Theta\Big) + \mathbf{y}^{(0)}, \quad l = 1, \dots, L, \\
\underbrace{\mathcal{P}(\boldsymbol{\xi})}_{\in \mathbb{R}^{N \times d}} &= \text{Projection}\Big(\mathbf{y}^{(L)}\Big).
\end{aligned}
\tag{2}
$$

The Adapter maps anchor features $\boldsymbol{\xi}$ to an initial prompt $\mathbf{y}^{(0)}$, which is recursively updated by $\mathcal{F}$ to $\mathbf{y}^{(l)}$ and, after $L$ steps, projected to the final prompt $\mathcal{P}(\boldsymbol{\xi})$. The operator $\mathcal{F}$ is anchor-specific: a two-layer MLP with GELU for $\mathcal{P}_{\text{in}}$ and a lightweight Conv1d block (kernels 3 and 1) for $\mathcal{P}_{\text{out}}$.

We adopt a tied-parameter recursion (Eq. (2)), a truncated fixed-point iteration inspired by Deep Equilibrium Models (DEQ) (Bai et al., 2019), which approximates the equilibrium state by solving:

$$g_{\Theta}(\mathbf{y}; \boldsymbol{\xi}) = \mathcal{F}(\mathbf{y}; \Theta) + \mathbf{y}^{(0)} - \mathbf{y} = 0. \tag{3}$$

If the refinement operator $\mathcal{F}(\cdot; \Theta)$ is a contraction, the recursion in Eq. (2) converges to a unique fixed point $y^{\star}$ with error $\|y^{(l)} - y^{\star}\| = \mathcal{O}(L^l)$ (Banach fixed-point theorem). In practice, we truncate at $L = 8$ iterations for a stable approximation, as validated in Fig. 3.

This recursive design directly addresses the limited expressivity of shallow anchors while avoiding the inefficiency of naive stacking. Concretely, each refinement step can be viewed as reprojecting anchors into the backbone's residual space, forming ***depth-equivalent*** virtual layers under fixed model size. Recursion further serves as a ***compute-only scaling knob***: increasing rounds extends inference depth without enlarging the model. Such behavior aligns with recent scaling-law findings (Alabdulmohsin & Zhai, 2025; Wang et al., 2024b), providing a principled strategy for ***scaling prompt adaptation within a single layer***. It also retains the constant-memory benefit of implicit-depth models, since no intermediate states need to be stored across rounds.

**(ii) *Residual injection.*** Once recursive prompts are generated, we integrate them into the backbone via residual blending. Direct replacement harms frozen features, whereas gated residual blending preserves stability. Given an input image $\mathbf{I}$, the patch embedding produces tokens $\eta \in \mathbb{R}^{N \times d}$. The input-side block modulates these tokens as:

$$\tilde{\eta} = (1 - \beta_1)\,\eta + \beta_1\,\mathcal{P}_{\text{in}}(\eta), \quad \beta_1 = \sigma\big(g_1(\eta)\big), \tag{4}$$

where $g_1(\eta)$ is a scalar gating logit from global average pooling followed by a 2-layer MLP, and $\sigma$ is the sigmoid. The frozen backbone $\mathcal{E}$ then produces intermediate features $z = \mathcal{E}(\tilde{\eta}) \in \mathbb{R}^{N' \times d'}$. Analogously, the output-side block is applied before the prediction head:

$$\tilde{z} = (1 - \beta_2)\,z + \beta_2\,\mathcal{P}_{\text{out}}(z), \quad \beta_2 = \sigma\big(g_2(z)\big). \tag{5}$$

Here $\beta_1, \beta_2 \in [0, 1]$ are dynamic gates that adaptively fuse original and prompt-modulated features, enabling residual blending.

**(iii) *Complementary dual prompts*.** We use two endpoint prompts: an input-side prompt ($\mathcal{P}_{\text{in}}$) that captures fine-grained spatial patterns from early features but tends to decay with depth (Yoo et al., 2023; Zhou et al., 2022a), and an output-side prompt ($\mathcal{P}_{\text{out}}$) that accesses high-level semantics from late features but is spatially coarse (Sun et al., 2024). This dual placement provides complementary adaptation that neither early nor late prompts alone can achieve.

Together with recursion and residual blending, it forms a lightweight, stable mechanism to modulate feature flow without disturbing the frozen backbone, thereby accommodating the structural variability of articulated objects.

## 3.3 Pyramid Representation Integration of Semantic and Geometry

Although prompt tuning alleviates the distribution gap from rigid pre-training, most existing category-level self-supervised methods (Liu et al., 2023b; Che et al., 2024) remain geometry-centric, leaving complementary semantic information largely unexploited. To this end, we propose the Cross Semantic–Geometry Pyramid ($\mathcal{X}$-SGP), which fuses semantic and geometric tokens into articulation-aware representations, resolving ambiguities under occlusion and symmetry.

**Semantic and Geometric Feature Extraction.** Given an input image $\mathbf{I}$, a frozen visual encoder $\Phi_V$ extracts semantic context, while a prompt-tuned encoder $\Phi_G^{\text{PT}}$ captures geometry-aware structure:

$$\mathcal{H}^{\text{img}} = \Phi_V(\mathbf{I}) \in \mathbb{R}^{N \times D}, \quad \boldsymbol{\Upsilon}^{\text{geo}} = \Phi_G^{\text{PT}}(\mathbf{I}) \in \mathbb{R}^{M \times D}, \tag{6}$$

where $\mathcal{H}^{\text{img}}$ denotes the semantic tokens and $\boldsymbol{\Upsilon}^{\text{geo}}$ denotes the geometric tokens.

**Architecture of $\mathcal{X}$-SGP.** As shown in Fig. 1(C), the geometry queries attend to semantic tokens, refining geometry features with semantic context to enhance the localization of articulated objects:

$$\widetilde{\boldsymbol{\Upsilon}}^{\text{geo}} = \text{CrossAttn}(\boldsymbol{\Upsilon}^{\text{geo}}_{\text{query}}, \mathcal{H}^{\text{img}}_{\text{key}}, \mathcal{H}^{\text{img}}_{\text{value}}). \tag{7}$$

Then, the semantic context is used to generate FiLM (Perez et al., 2018) parameters $\boldsymbol{\gamma}, \boldsymbol{\rho}$, which serve as scale and shift factors for feature-wise modulation:

$$\begin{aligned} [\boldsymbol{\gamma}, \boldsymbol{\rho}] &= \text{MLP}\big(\text{MeanPool}(\mathcal{H}^{\text{img}})\big), \\ \boldsymbol{\kappa}^{\text{raw}}_{\text{fused}} &= \text{ReLU}\big(\widetilde{\boldsymbol{\Upsilon}}^{\text{geo}} \odot (1 + \boldsymbol{\gamma}) + \boldsymbol{\rho}\big), \end{aligned} \tag{8}$$

where $\odot$ denotes element-wise multiplication; $\boldsymbol{\gamma}, \boldsymbol{\rho}$ modulate geometry features to stabilize alignment. The fused tokens are refined by a pyramid of DSConv (Howard et al., 2017). This refinement captures local interactions and articulation-induced deformations:

$$\boldsymbol{\kappa}^{\text{pyr}}_{\text{fused}} = \text{GELU}\Big(\boldsymbol{\kappa}^{\text{raw}}_{\text{fused}} + \big(\overset{\circ}{\prod}_{d \in \{1,2,4\}} \text{DSConv}_{3,d}\big)(\boldsymbol{\kappa}^{\text{raw}}_{\text{fused}})\Big), \tag{9}$$

where $\overset{\circ}{\prod}$ denotes sequential composition. This completes the $\mathcal{X}$-SGP module, which dynamically fuses semantic context and geometry features into articulation-aware representations (Implementation details are provided in Appendix C).

### 3.4 POSE REPRESENTATION AND ESTIMATION

**Point Cloud Prediction and Canonical Alignment.** Fused tokens $\kappa_{\text{fused}}^{\text{pyr}}$ are adapted by a lightweight MLP and passed through the frozen VGGT decoder to produce dense point clouds $\mathcal{Q}_{\text{pred}}$. These are then normalized to a canonical scale and translation, yielding $\widetilde{\mathcal{Q}}_{\text{norm}}$, which reduces global pose ambiguities (see Appendix D for details).

To ensure a consistent basis and remove global pose ambiguity, we normalize each point cloud and align it to a canonical representation by discretizing SE(3) into anchors and selecting the transformation that minimizes the Chamfer distance (CD) to a learnable category-level template:

$$(\mathbf{R}^*, \mathbf{t}^*) = \arg \min_{(\mathbf{R}_j, \mathbf{t}_j) \in \mathcal{A}} \text{CD}(\mathbf{R}_j \widetilde{\mathcal{Q}}_{\text{norm}} + \mathbf{t}_j, \mathcal{Q}_{\text{norm}}^{\text{ref}}). \tag{10}$$

The aligned point cloud is then $\widehat{\mathcal{Q}}_{\text{cano}} = \mathbf{R}^* \widetilde{\mathcal{Q}}_{\text{norm}} + \mathbf{t}^*$. This discrete alignment is applied only during training as an auxiliary supervision (see Appx. D.1 for normalization and discretization details).

**Physically-consistent Augmentation.** To alleviate scale ambiguity in monocular 3D reconstruction, we adopt a simple augmentation strategy that jointly perturbs the predicted canonicalized point cloud $\widehat{\mathcal{Q}}_{\text{cano}}^{\text{aug}}$, which promotes scale-invariant learning (see Appendix D.2 for details).

**Articulated Object Pose Estimation.** We feed the fused tokens $\kappa_{\text{fused}}^{\text{pyr}}$ into a unified `PoseHead`, following OP-Align (Che et al., 2024), which simultaneously handles object-level and part-level pose estimation. At the object level, `PoseHead` predicts $O$ candidate global transformations $\{(\mathbf{R}_o, \mathbf{t}_o)\}_{o=1}^O$:

$$\{(\mathbf{R}_o, \mathbf{t}_o)\}_{o=1}^O = \text{PoseHead}(\kappa_{\text{fused}}^{\text{pyr}}), \tag{11}$$

and selects the best candidate using nearest-neighbor distance (NND) scoring:

$$(\mathbf{R}_o, \mathbf{t}_o) = \arg \max_i -\text{NND}(\mathbf{R}_{[i]} \widehat{\mathcal{Q}}_{\text{cano}}^{\text{aug}} + \mathbf{t}_{[i]}, \mathcal{Q}_{\text{norm}}^{\text{ref}}). \tag{12}$$

At the part level, `PoseHead` regresses per-part rigid transformations together with articulation parameters $\{(T_k, \theta_k)\}_{k=1}^K$. The canonicalized global pose $(\mathbf{R}_o, \mathbf{t}_o)$ reduces variance, while the articulation parameters $\theta_k$ capture kinematic states such as joint rotations or prismatic displacements. Canonicalized supervision reduces pose variance. Meanwhile, semantic–geometric fusion ensures consistent alignment and accurate articulation recovery across diverse objects.

### 3.5 LOSS FUNCTION

We follow OP-Align (Che et al., 2024) and adopt its object- and part-level reconstruction losses, denoted as $\mathcal{L}_{\text{pose}}$. On top of this baseline, we introduce a DEQ-style regularization term with weight $\lambda$ to stabilize recursive prompt updates by encouraging fixed-point consistency (details in Appx. E)

$$\mathcal{L}_{\text{total}} = \underbrace{\mathcal{L}_{\text{pose}}}_{\text{Pose Estimation}} + \lambda \sum_{a \in \{\text{in, out}\}} \underbrace{\left\| \mathcal{F}_a(\mathbf{y}_a^{(L)}; \Theta) + \mathbf{y}_a^{(0)} - \mathbf{y}_a^{(L)} \right\|_F^2}_{\text{Recursive Residual Prompting}}. \tag{13}$$

## 4 EXPERIMENTS

In this section, we evaluate ReCAP on three articulated object benchmarks: two ***real-world*** datasets (OP-Align (Che et al., 2024), HOI4D (Liu et al., 2022b)) and one ***synthetic*** dataset (PartNet-Mobility (Xiang et al., 2020)). OP-Align and HOI4D cover scanned or human–object interaction scenarios, while PartNet-Mobility provides CAD-based articulated objects for controlled evaluation. (Additional dataset statistics and implementation details are provided in Appendix F.)

### 4.1 DATASETS AND EVALUATION METRIC

**OP-Align (*real-world*).** The OP-Align dataset (Che et al., 2024) is a scanned articulated-object benchmark with part-level pose annotations. We follow the standard protocol and evaluate on four categories: laptop, suitcase, drawer, and scissors. For part segmentation, we report mean Intersection-over-Union (mIoU) across all object parts. For pose evaluation, we follow the category-level 6D pose protocol (Che et al., 2024; Liu et al., 2023b), reporting average precision (AP) where

Table 1: Comparison with state-of-the-art methods on the real-world OP-Align dataset (Che et al., 2024). Evaluated articulated object categories include laptop, suitcase, drawer, and scissors. "**Supervised**" indicates that the method is trained with annotated RGB-D data, while "**self-supervised**" indicates that it is trained without any manual annotations. Our method **ReCAP** relies only on a **single RGB image** in a self-supervised manner. (↑: higher is better)

| Category | Type | Method | Modalities | Segm. (IoU↑) | | Joint Precision (AP↑) | | | Part Precision (AP↑) | | |
|---|---|---|---|---|---|---|---|---|---|---|---|
| | | | | 75% | 50% | 5°5cm | 10°10cm | 15°15cm | 5°5cm | 10°10cm | 15°15cm |
| Laptop | Supervised | 3DGCN (Lin et al., 2020) | RGB + Depth | 93.20 | 95.87 | 73.30 | 96.12 | 96.60 | 31.31 | 72.57 | 82.03 |
| | | EAP (Liu et al., 2023b) | RGB + Depth | 89.82 | 91.47 | 1.24 | 88.60 | 90.89 | 26.54 | 70.65 | 81.49 |
| | Self-supervised | OP-Align (Che et al., 2024) | RGB + Depth | 95.02 | 96.89 | 2.63 | 95.44 | 96.07 | 31.57 | 76.22 | 89.87 |
| | | **ReCAP** (Ours) | RGB | **96.39** | **97.70** | **5.21** | **96.16** | **97.25** | **33.12** | **77.08** | **90.63** |
| Suitcase | Supervised | 3DGCN (Lin et al., 2020) | RGB + Depth | 97.36 | 98.95 | 43.31 | 86.09 | 94.49 | 10.24 | 49.87 | 77.17 |
| | | EAP (Liu et al., 2023b) | RGB + Depth | 4.57 | 48.74 | 2.32 | 66.65 | 86.76 | 0.54 | 14.23 | 48.14 |
| | Self-supervised | OP-Align (Che et al., 2024) | RGB + Depth | 8.98 | 55.23 | 4.01 | 71.42 | 93.02 | 1.01 | 17.19 | 53.67 |
| | | **ReCAP** (Ours) | RGB | **11.70** | **57.89** | **6.43** | **73.57** | **94.29** | **2.96** | **19.40** | **56.29** |
| Drawer | Supervised | 3DGCN (Lin et al., 2020) | RGB + Depth | 82.52 | 96.90 | 38.27 | 76.32 | 91.15 | 20.80 | 68.36 | 88.50 |
| | | EAP (Liu et al., 2023b) | RGB + Depth | 1.24 | 49.02 | 4.03 | 47.65 | 76.28 | 4.73 | 65.45 | 75.73 |
| | Self-supervised | OP-Align (Che et al., 2024) | RGB + Depth | 3.11 | 53.87 | 9.26 | 54.79 | 81.42 | 6.98 | 70.02 | 80.44 |
| | | **ReCAP** (Ours) | RGB | **5.64** | **57.05** | **11.15** | **56.53** | **83.12** | **7.56** | **72.17** | **82.13** |
| Scissors | Supervised | 3DGCN (Lin et al., 2020) | RGB + Depth | 76.01 | 94.54 | 43.94 | 85.99 | 97.15 | 1.66 | 22.33 | 50.83 |
| | | EAP (Liu et al., 2023b) | RGB + Depth | 6.46 | 39.64 | 41.02 | 90.24 | 92.81 | 28.46 | 60.57 | 65.54 |
| | Self-supervised | OP-Align (Che et al., 2024) | RGB + Depth | 11.03 | 44.98 | 46.27 | 95.68 | **98.92** | 33.77 | 65.26 | 71.62 |
| | | **ReCAP** (Ours) | RGB | **13.17** | **46.92** | **48.09** | **96.10** | 98.64 | **35.57** | **67.19** | **72.84** |
| Avg. | Supervised | 3DGCN (Lin et al., 2020) | RGB + Depth | 87.27 | 96.56 | 49.70 | 86.13 | 94.84 | 16.00 | 53.28 | 74.63 |
| | | EAP (Liu et al., 2023b) | RGB + Depth | 25.52 | 57.22 | 12.15 | 73.29 | 86.69 | 15.07 | 52.73 | 67.73 |
| | Self-supervised | OP-Align (Che et al., 2024) | RGB + Depth | 29.54 | 62.74 | 15.54 | 79.33 | 92.36 | 18.33 | 57.17 | 73.90 |
| | | **ReCAP** (Ours) | RGB | **31.73** | **64.89** | **17.72** | **80.59** | **93.33** | **19.81** | **58.96** | **75.47** |

a prediction is correct if translation and rotation errors of all parts fall within 5/10/15 cm and $5°/10°/15°$, respectively. We further measure joint pivot and direction accuracy under the same thresholds, and report part segmentation mIoU at 75% and 50% overlap.

**HOI4D (*real-world*).** The HOI4D dataset (Liu et al., 2022b) is a large-scale 4D egocentric benchmark for category-level human–object interaction. Following the preprocessing of (Liu et al., 2023b), we evaluate on three categories: laptop, safe, and scissors. We report degree error (°) for 3D rotation, distance error (m) for 3D translation, angular error (°) for joint direction, and Chamfer Distance (CD-$\ell_1$) between predicted reconstruction and input partial point cloud for reconstruction quality.

**PartNet-Mobility (*synthetic*).** The PartNet-Mobility dataset (Xiang et al., 2020) is a CAD-based synthetic benchmark of articulated objects. We follow the preprocessing of (Xue et al., 2021) and evaluate on six categories: laptop, drawer, scissors, eyeglasses, dishwasher, and safe. We report results on both clean and randomly occluded partial image inputs, measuring performance by mean Intersection-over-Union (mIoU) across parts and inference speed in frames per second (FPS).

It is worth noting that current category-level datasets (Che et al., 2024; Liu et al., 2022b; Xiang et al., 2020) exhibit a clear imbalance between rotational and prismatic articulations. While rotational categories (e.g., laptops, suitcases) dominate most datasets, prismatic ones (e.g., drawers, safes) appear only sparsely. This distributional bias mirrors the practical constraints of data collection—prismatic motions often cause severe self-occlusion and partial visibility—yet it also highlights the need for future benchmarks to include a more balanced articulation taxonomy for fairer evaluation.

### 4.2 RESULTS ON BENCHMARK DATASETS

We compare with 3DGCN (Lin et al., 2020), EAP (Liu et al., 2023b), and OP-Align (Che et al., 2024), which are evaluated under their original ***RGB-D*** input setting, while ReCAP uses only a ***single RGB image***, highlighting stronger generalization in the more challenging RGB-only regime.

**Results on OP-Align dataset.** As shown in Table 1, ReCAP in the self-supervised single-image setting consistently outperforms all self-supervised baselines, and even surpasses the supervised RGB-D method 3DGCN (Lin et al., 2020) on several categories. On average, it improves part segmentation mIoU by **+2.15** and joint/part AP by **+1–2** points, achieving the best performance

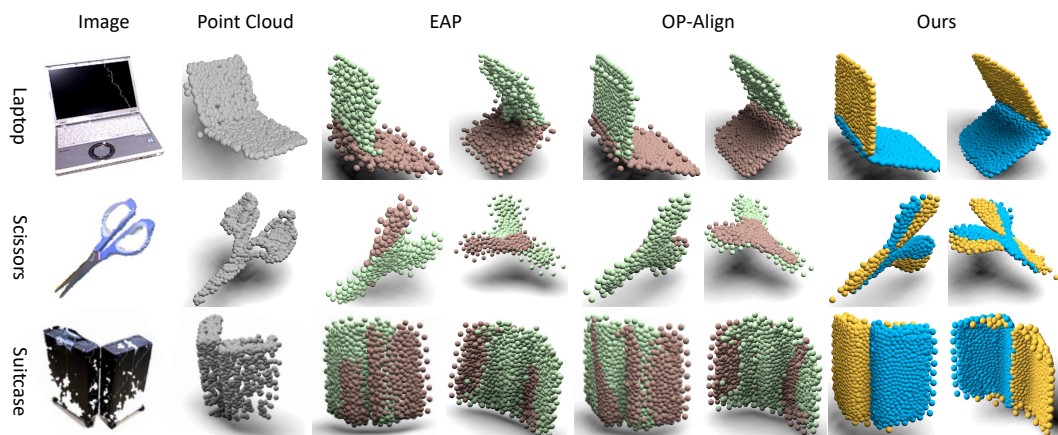

Figure 2: Qualitative comparison on object pose estimation from segmented reconstructions. Our method takes a single RGB image as input, while EAP and OP-Align use depth-scanned point clouds. Two views are shown for each method, with segmentation indicated by color to illustrate part-segmented reconstruction quality and accurate articulation.

on challenging objects such as Laptop and Scissors. These gains demonstrate the effectiveness of Recursive Residual Prompting in transferring rigid-object priors to articulated settings, while $\mathcal{X}$-SGP resolves ambiguities from symmetry and occlusion.

For qualitative comparison, Fig. 2 visualizes input images, scanned point clouds, and reconstructions from EAP, OP-Align, and ReCAP. While the baselines rely on depth-scanned point clouds, our method requires only a single RGB image. ReCAP produces reconstructions with clear part segmentation and accurate articulation, faithfully reflecting the object's observed pose. On suitcases (third row), it preserves sharp hinge boundaries and distinctly separates the two flaps while capturing their opening states, in contrast to prior methods that yield blurred or ambiguous boundaries. Even without human labels, ReCAP surpasses prior self-supervised baselines and approaches supervised performance, establishing a new SoTA in real-world articulated pose estimation, highlighting the effectiveness of combining recursive prompting with foundation models and semantic cues.

**Results on HOI4D.** To further validate our approach in a distinct real-world scenario, we evaluate ReCAP on the HOI4D dataset (Liu et al., 2022b), which provides richer annotations and allows assessment of pose accuracy and reconstruction quality. As shown in Table 2, ReCAP consistently outperforms both self-supervised EAP and OP-Align baselines across rotation error, translation error, joint direction error, and reconstruction quality (CD-$\ell_1$). On average, ReCAP reduces rotation error by **0.33–5.61°** and joint direction error by over **30°**, while maintaining translation accuracy on par with OP-Align and further lowering reconstruction distance. These results demonstrate that ReCAP achieves substantial improvements without sacrificing other metrics, confirming its robustness and generalization in egocentric settings.

Table 2: Results on the real-world HOI4D dataset (Liu et al., 2022b). We report rotation error, translation error, joint direction error, and reconstruction quality (CD-$\ell_1$). ($\downarrow$ is better)

| | Method | Laptop | Safe | Scissors | Avg. |
|---|---|---|---|---|---|
| Rotation Error ($°\downarrow$) | EAP (Liu et al., 2023b) | 7.71 | 18.65 | 7.26 | 9.88 |
| | OP-Align (Che et al., 2024) | 4.27 | 6.54 | 5.98 | 4.60 |
| | **ReCAP (Ours)** | **3.98** | **5.81** | **5.02** | **4.27** |
| Translation Error ($m\downarrow$) | EAP (Liu et al., 2023b) | **0.079** | 0.065 | 0.102 | 0.082 |
| | OP-Align (Che et al., 2024) | 0.090 | 0.064 | **0.084** | **0.079** |
| | **ReCAP (Ours)** | 0.086 | **0.058** | 0.094 | 0.080 |
| Joint Direction ($°\downarrow$) | EAP (Liu et al., 2023b) | 18.02 | 55.16 | - | 36.59 |
| | OP-Align (Che et al., 2024) | 1.46 | 1.34 | - | 1.40 |
| | **ReCAP (Ours)** | **1.39** | **1.30** | - | **1.35** |
| Reconstruction (CD-$\ell_1 \times 100\downarrow$) | EAP (Liu et al., 2023b) | 3.28 | 3.62 | 3.15 | 3.35 |
| | OP-Align (Che et al., 2024) | 2.92 | 3.27 | **2.83** | 3.01 |
| | **ReCAP (Ours)** | **2.65** | **3.01** | 2.92 | **2.86** |

**Results on PartNet-Mobility.** Since real-world datasets are limited in scale, we also evaluate on the synthetic PartNet-Mobility dataset (Xiang et al., 2020) to assess generalization. As shown in Table 3, we report results under two input conditions: clean inputs (w/o Occ.) compared with SoTA baselines, and inputs with random occlusion masks (w/ Occ., blue) to simulate real-world scenarios.

Table 3: Results on the PartNet-Mobility (Xiang et al., 2020), reporting part segmentation (Segm. mIoU, %) over four articulated object categories. "Occ." denotes occlusion; "w/o" is the average on clean inputs, and "w/" on occluded ones (blue). "Speed" is inference throughput in FPS. (↑ is better)

| | Method | Laptop | Eyeglasses | Drawer | Scissors | Dishwasher | Safe | Avg. (w/o Occ.) / (w/ Occ.) | Speed (FPS ↑) |
|---|---|---|---|---|---|---|---|---|---|
| Segm. (IoU↑) | EAP (Liu et al., 2023b) | 79.25 | 58.79 | 49.36 | 38.23 | - | - | 56.40 | < 1 |
| | OP-Align (Che et al., 2024) | **82.92** | 62.18 | 51.22 | 41.49 | 79.87 | 68.61 | 64.39 | **41** |
| | **ReCAP** (Ours) | 80.64 / 72.36 | **64.41** / 52.71 | **53.08** / 45.57 | **44.15** / 31.94 | **82.18** / 71.02 | **69.46** / 60.27 | **65.64** / 55.65 | 13 |

ReCAP achieves the best overall performance (**65.64**% mIoU), showing strong results on thin or occluded parts such as Eyeglasses, Drawer, and Scissors, with only a slight drop on Laptop, likely due to the synthetic dataset's geometric bias and limited semantic cues. Results under occlusion (w/ Occ., blue) serve as a simulation of real-world scenarios, further confirming the robustness of our approach. Despite a lower FPS (13 vs. 41), ReCAP remains practically feasible, marking a key step toward deployment without requiring multi-view or depth inputs.

## 4.3 ABLATION STUDIES

Table 4: Ablation study on the complementary dual prompts. $\mathcal{P}_{in}$ and $\mathcal{P}_{out}$ denote prompt blocks inserted before patch embedding and after the prediction head, respectively. Results for baseline, single-prompt, and dual-prompt variants. Metrics are segmentation IoU, joint precision, and part precision (↑ is better).

| Method | | Segmentation | Joint Precision | Part Precision |
|---|---|---|---|---|
| $\mathcal{P}_{in}$ | $\mathcal{P}_{out}$ | (IoU↑) | (mAP↑) | (mAP↑) |
| (A) | | 30.52, 62.57 | 17.15, 78.49, 91.11 | 18.96, 57.32, 73.36 |
| (B) ✓ | | 31.02, 63.58 | 17.36, 79.47, 92.53 | 19.19, 58.07, 74.30 |
| (C) | ✓ | 31.09, 64.02 | 17.49, 79.96, 92.91 | 19.25, 58.16, 74.71 |
| (D) ✓ | ✓ | 31.73, 64.89 | 17.72, 80.59, 93.33 | 19.81, 58.96, 75.47 |

Table 5: Ablation of stacking layers (1–4) in the Recursive Residual Prompting on real-world and synthetic datasets. Both $\mathcal{P}_{in}$ (4.2M, black) and $\mathcal{P}_{out}$ (6.0M, blue) show limited gains with deeper stacking at the cost of more parameters.

| Layers | OP-Align | HIO4D | PartNet-Mobilty |
|---|---|---|---|
| | (IoU↑) | (CD-$\ell_1$↓) | (IoU↑) |
| 1 | 63.54, 63.89 | 2.93, 2.92 | 59.17, 59.61 |
| 2 | 63.60, 64.08 | 2.93, 2.91 | 59.24, 59.73 |
| 3 | 63.69, 64.14 | 2.92, 2.89 | **59.37**, **59.82** |
| 4 | **63.72**, **64.21** | **2.92**, **2.89** | 59.28, 59.70 |

**Impact of Complementary Dual Prompts.** Table 4 shows that both input-side ($\mathcal{P}_{in}$) and output-side ($\mathcal{P}_{out}$) prompts contribute complementary gains on the OP-Align dataset (Che et al., 2024). (A) Removing all prompts reduces the model to a frozen backbone baseline. (B) Using only $\mathcal{P}_{in}$ gives modest gains from low-level modulation, while (C) using only $\mathcal{P}_{out}$ yields larger gains by adapting high-level semantics. (D) Combining both achieves the best results across all metrics, confirming that complementary dual prompts are essential for robust articulated pose estimation.

**Impact Compared to Layer Stacking.** Table 5 shows that across all evaluated benchmarks, stacking 1–4 layers yields only minor gains for $\mathcal{P}_{in}$ (**4.2M**) and $\mathcal{P}_{out}$ (**6.0M**), while parameters and computation increase almost linearly. Since dual prompts already capture key semantic–geometric interactions, further stacking is largely redundant. In contrast, our single-layer recursive update achieves ***depth-equivalent adaptation*** with shared parameters, driving prompts toward a fixed-point equilibrium while updating only ∼10.2M parameters (∼0.8% of VGGT's 12.6B), showing that lightweight prompts alone are sufficient to unlock the representational capacity of the foundation model.

**Impact of Recursion Rounds.** As shown in Fig. 3, increasing the number of recursive refinement rounds $L$ improves segmentation mIoU@50 up to $L = 6$, with only marginal gains beyond $L = 8$ where the curve saturates. We therefore adopt $L = 8$ as the default to balance accuracy and inference latency. This trend confirms that recursion rounds yield DEQ-style depth-equivalent virtual layers under fixed parameter cost, serving as a ***compute-efficient knob*** for prompt refinement until performance saturates.

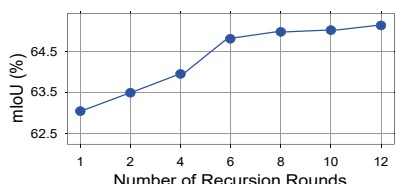

Figure 3: Recursion ablation on OP-Align dataset. (Segm. mIoU@50)

To further illustrate this effect, Fig. 4 visualizes segmentation evolution across recursion rounds. Recursive prompting progressively sharpens part boundaries and aligns local geometry into semantically consistent configurations. Early rounds produce coarse part grouping, while later rounds refine fine-grained boundaries and eliminate residual misalignment. This visual progression con-

firms that recursion acts as a depth-equivalent refinement mechanism, continuously improving the feature representations until convergence, thereby yielding more reliable part structure and directly benefiting articulated object pose estimation.

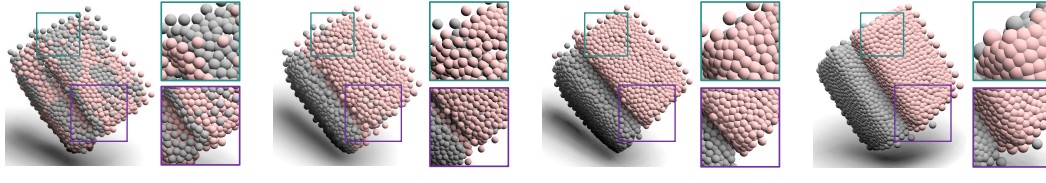

Recursion Round 1      Recursion Round 4      Recursion Round 8      Recursion Round 12

Figure 4: Segmentation progressively improves across recursion rounds. Each round denotes one re-finement iteration in our recursive prompting pipeline. As recursion proceeds ($1 \rightarrow 12$), part bound-aries sharpen and geometric alignment becomes more consistent, showing how recursive prompting incrementally refines semantic and geometric structure without extra supervision.

**Impact of $\mathcal{X}$-SGP.** Table 6 shows that geometry alone (A) performs worst, confirming the ambiguity of purely geometric cues. (B) brings only limited gains since it lacks adaptive multi-scale fusion, while removing FiLM (C) or the pyramid (D) further degrades cross-modal alignment and articulation modeling. These re-sults highlight the necessity of all components, with the full $\mathcal{X}$-SGP (E) achieving the strongest performance for articulation-aware representation learning.

Table 6: Ablation study on the Cross Semantic-Geometry Pyramid ($\mathcal{X}$-SGP).

| Method | Joint Precision (mAP↑) | Part Precision (mAP↑) |
|---|---|---|
| (A) Geom. Only | 14.62, 77.17, 89.23 | 16.28, 55.46, 72.05 |
| (B) w/ Concat | 16.28, 78.74, 91.19 | 18.67, 57.09, 73.64 |
| (C) w/o FiLM | 18.15, 80.02, 92.81 | 19.04, 58.12, 74.76 |
| (D) w/o Pyramid | 16.83, 79.43, 92.72 | 18.58, 57.24, 74.45 |
| (E) Full Model | 17.72, 80.59, 93.33 | 19.81, 58.96, 75.47 |

## 5   CONCLUSION

We present ReCAP, the first self-supervised framework for category-level articulated pose estimation from a single RGB image. It adapts the geometry foundation model to articulated targets through parameter-efficient recursive prompting and enhances robustness via semantic–geometry fusion. By bridging the gap between rigid-object priors and articulated variability, ReCAP overcomes the limi-tations of prior paradigms and paves the way for practical monocular articulated pose estimation.

**Limitation and Future Work.** While single-image estimation inherently lacks multi-view cues, our results demonstrate strong performance under this challenging setting. Future work will explore incorporating multi-view and richer contextual signals, and extend ReCAP toward object-agnostic articulated pose estimation in diverse real-world environments.

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

## APPENDIX

This supplementary document provides additional details to support the main paper.

## A    LLM USAGE

We disclose that ChatGPT (GPT-5) was used during manuscript preparation. Its role was limited to assisting with grammar correction and improving readability. No technical contributions, research ideas, experimental designs, or analyses were generated by an LLM. All scientific content and conclusions are solely the responsibility of the authors.

## B    THEORETICAL AND EMPIRICAL ANALYSIS

### B.1    ANALYSIS OF THE STATIC-TOPOLOGY VIOLATION

**Static Topology Assumption.** Conventional prompt tuning implicitly assumes that the token topology is static across layers, i.e., token indices remain fixed such that prompts can be consistently aligned with their corresponding embeddings. Formally,

$$[\mathbf{E}_i; \mathbf{P}_i] = f_i([\mathbf{E}_{i-1}; \mathbf{P}_{i-1}]), \quad i = 1, \ldots, N, \tag{14}$$

where $f_i$ denotes the $i$-th encoder block and $[\cdot; \cdot]$ indicates token-wise concatenation.

**VGGT Token Dynamics.** Unlike static-token models, VGGT reconfigures its token composition at each layer by assembling camera tokens $\mathbf{C}_i$, register tokens $\mathbf{R}_i$, and patch tokens $\mathbf{E}_i$. Formally,

$$\mathcal{T}_i = \mathtt{Arrange}_i(\mathbf{C}_i, \mathbf{R}_i, \mathbf{E}_i), \tag{15}$$

where $\mathtt{Arrange}_i(\cdot)$ alters both the cardinality and ordering of tokens at each layer, thereby disrupting the fixed positional correspondence assumed in Eq. (14).

**Violation.** Since the token arrangement varies across layers, no consistent mapping exists to preserve the one-to-one alignment between prompts and embeddings. Formally,

$$\nexists \phi \quad [\mathcal{T}_i; \mathbf{P}_i] = f_i([\phi(\mathcal{T}_{i-1}); \mathbf{P}_{i-1}]), \tag{16}$$

i.e., no fixed positional mapping $\phi$ exists that preserves prompt alignment under dynamic token rearrangement.

This incompatibility is not unique to VGGT, but a general limitation of applying conventional prompt tuning to hierarchical or dynamic-token architectures (e.g., token pruning, multi-view backbones). This formally explains why conventional prompt tuning yields limited gains when applied to VGGT (see Sec. 3.2).

### B.2 Supplementary Experiments

Table 7: Supplementary comparison of VGGT with and without shallow prompt tuning on the OP-Align dataset, evaluated by segmentation (IoU), joint precision (mAP), and part precision (mAP).

| Method | Segmentation (IoU↑) | Joint Precision (mAP↑) | Part Precision (mAP↑) |
|---|---|---|---|
| w/o prompt | 30.52, 62.57 | 17.15, 78.49, 91.11 | 18.96, 57.32, 73.36 |
| w/ prompt | 30.70, 63.07 | 17.17, 79.01, 91.82 | 19.21, 57.75, 73.62 |

As shown in Table 7, on the OP-Align dataset, directly prepending shallow prompts to VGGT yields negligible gains across all metrics. The improvements are marginal ($<$0.7 mAP) and within typical run-to-run variation, suggesting that prompts fail to exert stable influence on VGGT. This phenomenon substantiates our claim of dynamic token misalignment: in VGGT, token sets are reorganized layer by layer, breaking the static index correspondence assumed in Eq. (1). Consequently, prepended prompts lose consistent alignment with task tokens, which explains their ineffectiveness despite adding extra parameters.

## C Implementation Details of $\mathcal{X}$-SGP

We provide additional details of the Cross Semantic–Geometry Pyramid ($\mathcal{X}$-SGP) introduced in Sec. 3.3, including examples of semantic and geometric tokens, parameterization of FiLM modulation, and the design of the multi-scale refinement module.

**Semantic and Geometric Examples.** For semantics, we adopt a frozen DINOv2 ViT-B/16 encoder to extract $\mathcal{H}^{\text{img}} \in \mathbb{R}^{N \times D}$, where $D = 768$ is the embedding dimension of the backbone and $N$ is the number of visual tokens produced by the encoder. We empirically observe that these tokens often correspond to object parts such as laptop lids, scissor blades, suitcase flaps, and drawer panels. For geometry, $\boldsymbol{\Upsilon}^{\text{geo}} \in \mathbb{R}^{M \times D}$ is obtained from the prompt-tuned VGGT encoder. These two streams are later fused in $\mathcal{X}$-SGP (Sec. 3.3).

**FiLM Parameters.** For clarity, the modulation in Eq. (8) uses FiLM parameters $(\boldsymbol{\gamma}, \boldsymbol{\rho}) \in \mathbb{R}^D$ derived from semantic context. We obtain a global descriptor $\bar{h} = \text{MeanPool}(\mathcal{H}^{\text{img}}) \in \mathbb{R}^D$, normalize it with LayerNorm, and map it through a two-layer MLP:

$$[\boldsymbol{\gamma}, \boldsymbol{\rho}] = W_2\, \phi(W_1\, \text{LN}(\bar{h})) \in \mathbb{R}^{2D}, \tag{17}$$

where $\phi$ is GELU. These parameters are broadcast to geometry tokens $\widetilde{\boldsymbol{\Upsilon}}^{\text{geo}} \in \mathbb{R}^{M \times D}$ as

$$\boldsymbol{\kappa}^{\text{raw}}_{\text{fused}} = \sigma\big(\widetilde{\boldsymbol{\Upsilon}}^{\text{geo}} \odot (1 + \boldsymbol{\gamma}) + \boldsymbol{\rho}\big), \tag{18}$$

with $\sigma$ denoting ReLU. For stability, the last projection layer is zero-initialized so that the modulation starts from the identity mapping. $(\boldsymbol{\gamma}, \boldsymbol{\rho})$ act as feature-wise **scaling** and **shifting** factors applied to the geometry stream. This semantic-driven modulation stabilizes cross-modal alignment and highlights articulation-specific variations that are often ambiguous when relying on geometry alone.

**Multi-Scale Refinement.** The refinement module in Eq. (19) is implemented as a pyramid of depthwise separable 1D convolutions. Each DSConv layer consists of a depthwise convolution (groups $= C$), a pointwise $1 \times 1$ convolution, BatchNorm, and a GELU activation. We use kernel size 3 and dilation rates $\{1, 2, 4\}$, with padding chosen to preserve sequence length. The three DSConv (Howard et al., 2017) layers are applied sequentially:

$$\big(\overset{\circ}{\prod}_{d \in \{1,2,4\}} \text{DSConv}_{3,d}\big)(x) = \text{DSConv}_{3,4}(\text{DSConv}_{3,2}(\text{DSConv}_{3,1}(x)))\,. \tag{19}$$

The output of this pyramid is added back to the input (residual connection) and passed through a GELU, as shown in Eq. (9) of the main paper.

$\mathcal{X}$-SGP integrates semantic cues with geometric features through FiLM-based modulation and multi-scale refinement. Semantic information provides high-level context that guides geometry tokens via feature-wise scaling and shifting, while the pyramid captures local deformation patterns across different receptive fields. Together, these mechanisms mitigate ambiguities from symmetry and occlusion and substantially improve part-level reasoning for articulated object pose estimation.

# D  CANONICALIZATION AND AUGMENTATION

## D.1  POINT CLOUD PREDICTION AND CANONICALIZATION

**Latent Prompt-based Point Cloud Prediction.** With the fused features $\boldsymbol{\kappa}_{\text{fused}}^{\text{pyr}}$ from the prompt-tuned VGGT, we add a lightweight MLP head $\mathcal{E}_{\text{pred}}$ that projects them into the representation space required by VGGT's decoder $\Phi_G^{\text{D}}$, which then outputs the dense 3D point cloud $\mathcal{Q}_{\text{pred}}$:

$$\mathcal{Q}_{\text{pred}} = \Phi_G^{\text{D}}\Big(\mathcal{E}_{\text{pred}}(\boldsymbol{\kappa}_{\text{fused}}^{\text{pyr}})\Big), \quad \mathcal{Q}_{\text{pred}} \in \mathbb{R}^{H \times W \times 3}. \tag{20}$$

The predicted point clouds are then aligned to a canonical coordinate frame, removing pose ambiguities and providing a consistent geometric basis for downstream pose estimation.

**Instance Normalization and Canonical Alignment.** To provide consistent supervision and reduce geometric variance, we normalize and canonicalize predicted point clouds $\mathcal{Q}_{\text{pred}}$. First, instance-level normalization centers each point cloud at its centroid $\boldsymbol{\mu}$ and rescales it by the mean distance $d$ to the centroid, with an additional fixed factor $s_0$ to unify global scale:

$$\widetilde{\mathcal{Q}}_{\text{norm}} = \frac{\mathcal{Q}_{\text{pred}} - \boldsymbol{\mu}}{d} \cdot s_0. \tag{21}$$

where $\widetilde{\mathcal{Q}}_{\text{norm}}$ denotes the normalized point coordinates.

Next, to obtain a canonical representation, we follow (Che et al., 2024) by discretizing the SE(3) group into a finite set of candidate transformations (anchors). For each anchor $(\mathbf{R}_j, \mathbf{t}_j) \in \mathcal{A}$, we transform the normalized point cloud and compute the Chamfer distance (CD) to a learnable category-level template $\mathcal{Q}_{\text{norm}}^{\text{ref}}$:

$$(\mathbf{R}^*, \mathbf{t}^*) = \arg \min_{(\mathbf{R}_j, \mathbf{t}_j) \in \mathcal{A}} \text{CD}(\mathbf{R}_j \widetilde{\mathcal{Q}}_{\text{norm}} + \mathbf{t}_j, \mathcal{Q}_{\text{norm}}^{\text{ref}}), \tag{22}$$

where $\text{CD}(\cdot, \cdot)$ is the symmetric Chamfer Distance:

$$\text{CD}(P, Q) = \frac{1}{|P|} \sum_{p \in P} \min_{q \in Q} \|p - q\|_2^2 + \frac{1}{|Q|} \sum_{q \in Q} \min_{p \in P} \|p - q\|_2^2. \tag{23}$$

The aligned point cloud is then given by $\widehat{\mathcal{Q}}_{\text{cano}} = \mathbf{R}^* \widetilde{\mathcal{Q}}_{\text{norm}} + \mathbf{t}^*$. This discrete SE(3) alignment is applied only during training, serving as auxiliary supervision to enforce canonical geometry and category-level consistency across instances. This prevents the network from wasting capacity on trivial global pose variations, allowing it to focus on articulated part structure.

## D.2  PHYSICALLY-CONSISTENT AUGMENTATION

Monocular dense reconstruction suffers from inherent scale ambiguity—especially in large vision models such as VGGT—which poses a fundamental obstacle to robust, category-level pose reasoning. Existing augmentations such as OP-Align (Che et al., 2024) alleviate rotation variance via global rotations and local perturbations, but overlook systematic scale discrepancies between predicted and canonical geometries.

To close this gap, we extend OP-Align by retaining its global rotation $\mathbf{R} \in \text{SO}(3)$ and Gaussian noise, and further introduce a uniform instance-level scaling applied consistently to both predicted point clouds and all associated labels. This physically-consistent augmentation ensures supervision remains aligned under arbitrary global scaling, mitigating scale-induced artifacts and improving generalization to objects of varying sizes and out-of-distribution scales.

Concretely, given the canonicalized object- and part-level point clouds $\widehat{\mathcal{Q}}_{\text{cano}}$ and $\widehat{\mathcal{Q}}_{\text{cano},k}$, the canonical joint axis $\mathbf{a}_k$ and part transform $(\mathbf{R}_k, \mathbf{t}_k)$ (w.r.t. the canonical/root frame, where $k$ indexes the $k$-th part), we sample $s_{\text{aug}} \in [0.9, 1.1]$ and apply

$$
\widehat{\mathcal{Q}}_{\text{cano}}^{\text{aug}} = s_{\text{aug}} \mathbf{R} \widehat{\mathcal{Q}}_{\text{cano}}, \qquad \widehat{\mathcal{Q}}_{\text{cano},k}^{\text{aug}} = s_{\text{aug}} \mathbf{R} \widehat{\mathcal{Q}}_{\text{cano},k},
$$
$$
\mathbf{a}_k^{\text{aug}} = \mathbf{R} \mathbf{a}_k, \qquad \mathbf{T}_k^{\text{aug}} = \begin{bmatrix} \mathbf{R}\mathbf{R}_k & s_{\text{aug}} \mathbf{R}\mathbf{t}_k \\ \mathbf{0}^\top & 1 \end{bmatrix}. \tag{24}
$$

Here rotations act on points/axes by left-multiplication (consistent with $\mathbf{R}_j \widetilde{\mathcal{Q}}_{\text{norm}} + \mathbf{t}_j$ in Sec. D.1), and the translation component scales with $s_{\text{aug}}$ to preserve physical consistency under uniform scaling. This regularizes scale inconsistencies in RGB-only pipelines and encourages the model to learn scale-invariant features critical for category-level articulated pose reasoning.

# E   LOSS FUNCTION DETAILS

## E.1   BASELINE POSE LOSS FROM OP-ALIGN

We adopt the object- and part-level reconstruction losses of OP-Align (Che et al., 2024), denoted as $\mathcal{L}_{\text{pose}}$. For completeness, we provide the original definitions here.

**Object-level Loss.**   The object-level reconstruction is optimized via bi-directional Density-aware Chamfer Distance (DCD):

$$
\mathcal{L}_o = \text{DCD}(R_o X + t_o, Y, 1, \alpha_L) + \text{DCD}(Y, R_o X + t_o, 1, \alpha_R), \tag{25}
$$

where $\alpha_L = 30$ and $\alpha_R = 120$ are temperature parameters.

**Part-level Loss.**   The part-level reconstruction employs segmentation-weighted DCD:

$$
\mathcal{L}_p = \sum_{j=1}^{P-1} \sum_{i=1}^{2} \frac{1}{b[j,i]} \Big( \text{DCD}(Z[j,i], Y, W_x[\sigma(j,i)], \alpha_L) + \text{DCD}(Y, Z[j,i], W_y[\sigma(j,i)], \alpha_R) \Big). \tag{26}
$$

**Additional Regularization.**   Following OP-Align, the full pose loss is defined as

$$
\mathcal{L}_{\text{pose}} = \mathcal{L}_o + \mathcal{L}_p + \mathcal{L}_{\text{reg}S} + \mathcal{L}_{\text{reg}D} + \mathcal{L}_{\text{reg}W} + \mathcal{L}_{\text{reg}P} + \mathcal{L}_{\text{reg}A} + \mathcal{L}_{\text{reg}J}, \tag{27}
$$

where the terms respectively enforce shape stability, density regularization, segmentation balance, shared-part consistency, canonical joint state, and pivot regularization (see (Che et al., 2024) for details).

## E.2   TOTAL LOSS IN OUR METHOD

In the main paper, we build on $\mathcal{L}_{\text{pose}}$ and add a DEQ-style fixed-point regularization to stabilize recursive prompt updates:

$$
\mathcal{L}_{\text{total}} = \mathcal{L}_{\text{pose}} + \lambda \sum_{a \in \{\text{in, out}\}} \left\| \mathcal{F}_a\big(\mathbf{y}_a^{(L)}; \Theta\big) + \mathbf{y}_a^{(0)} - \mathbf{y}_a^{(L)} \right\|_F^2. \tag{28}
$$

# F   DATASET DETAILS

**OP-Align Dataset.** The official OP-Align release provides dense color images and auxiliary annotations in flattened arrays of size $480 \times 640$. We reconstruct the original image format by reshaping these arrays, apply foreground cropping with the provided segmentation masks, and resize all images to $518 \times 518$ to match the input resolution of VGGT. This ensures consistent alignment between appearance and geometry modalities throughout training and evaluation.

**PartNet-Mobility Dataset.** We follow OMAD (Xue et al., 2021) to process the PartNet-Mobility dataset into synthetic RGB renderings with per-part segmentation for evaluation. The official training and testing splits are used without further modifications.

# G IMPLEMENTATION DETAILS

## G.1 EVALUATION SETTINGS

Following the official EAP protocol (Liu et al., 2023b), we compose predicted part transformations (relative to the canonical/root frame) with the benchmark reference pose to align coordinate systems for evaluation. For symmetric objects, we use the OP-Align protocol, which exhaustively enumerates valid part permutations and reports the best mIoU.

## G.2 TRAINING AND IMPLEMENTATION

All input RGB images are resized to $518 \times 518$, following the official VGGT setting and pre-trained weight release to ensure compatibility and reproduce the reported performance. This resolution is critical for three reasons: (1) VGGT was pretrained on square inputs, and preserving this resolution maximizes geometric fidelity in predicted point clouds $\mathcal{Q}_{\text{pred}} \in \mathbb{R}^{H \times W \times 3}$; (2) center-padding retains all original pixels, avoiding distortions that occur with cropping or aspect-ratio preserving padding; (3) prompt tuning operates on frozen VGGT weights, so strict consistency with the pretraining input resolution is required.

All predicted point clouds are uniformly downsampled to 2,048 points for efficient training and evaluation. We train all models for 350 epochs on an NVIDIA H100 GPU with 80GB memory.

