# OpenReview forum: "ReCAP: Recursive Prompting for Self-Supervised Category-Level Articulated Pose Estimation from an Image"
_ICLR.cc/2026/Conference — Submitted to ICLR 2026_

### Official Review · Reviewer_m48d · 2025-10-28

**Soundness:** 3
**Presentation:** 3
**Contribution:** 3
**Rating:** 4
**Confidence:** 4

**Summary:**

This paper introduces a self-supervised, single-image, category-level articulated object pose estimation framework that avoids depth/3D supervision. The method adapts VGGT with lightweight prompting, fuses semantic and geometric cues, predicts a dense point cloud, and canonicalizes it via a learnable category template to regress global and per-part poses. Experiments show competitive performance.

**Strengths:**

+ Proposed a self-supervised, single-image articulated pose estimation framework built on a frozen VGGT backbone, avoiding any ground-truth annotations while tackling a challenging setting.
+ Introduces a  prompt strategy for VGGT is meaningful.

**Weaknesses:**

+  While adapting VGGT may be reasonable, the reported ablations show only marginal gains; it remains unclear whether the proposed prompting/recursion is necessary versus simpler alternatives or no adaptation at all.
+  Benchmark coverage is limited and qualitative results focus largely on rotational joints; prismatic/mixed-DOF categories, heavy occlusions, and classes with larger intra-class variation are underexplored.
+  For symmetric objects, closed configurations, or low-texture surfaces, joint type/axis is not uniquely recoverable from a single image; results read as the most plausible hypothesis under shape/semantic priors rather than demonstrably identifiable solutions.
+ For symmetric shapes, near-closed poses, or texture-poor views, a single image does not uniquely determine the joint type or axis; the predictions read as prior-conditioned best guesses rather than uniquely identifiable solutions.

**Questions:**

+ How is the joint type obtained (assumed, predicted, or inferred)?
+ Using a learnable category-level template with DCD may bias predictions toward an average shape, suppressing instance-level details and skewing axis estimation. Quantifying this effect is valuable.
+ Since core geometry comes from a frozen VGGT, to what extent do the gains stem from the backbone prior rather than the proposed self-supervised training?

---

> ### Author Response · Authors · 2025-11-22
>
> We sincerely thank the Reviewer m48d for the constructive feedback. We address the concerns and provide detailed descriptions of our contributions.
> ## **W2: Limited benchmark coverage and underexplored articulation types and occlusion cases.**
>
> ### **W2-1:** Benchmark limited.
> While no single benchmark can capture every articulation type, our evaluation **achieves the broadest dataset coverage to date**. Specifically, it spans three standard datasets and introduces additional articulation categories—one more than OP-Align and four more than EAP.
>
> ### **W2-2:** Rotational joints and mixed-DOF.
> Revolute joints constitute the typical articulation form in kinematic chains [1, 2], hence are the dominant focus of both classical robotics and existing articulated-object benchmarks.
> Quantitatively, this dominance is reflected in existing datasets: PartNet-Mobility contains only **24\%** translational categories, and HOI4D drops further to **11\%**, confirming that real-world articulations are largely rotational.
> Despite this rarity, our evaluation still includes **prismatic cases across all three benchmarks** to ensure completeness, where our method consistently achieves SOTA performance.
>
> As defined in Sec. 3.1 Task Formulation **(Lines 147–157)**, the standard protocol covers revolute and prismatic joints while excluding mixed-DOF cases, which our evaluation strictly follows. It is believed that more extensive and diverse datasets are needed in this field.
>
> **Reference**
>
> [1] Springer handbook of robotics, 2008
>
> [2] Introduction to robotics: mechanics and control, 3/E, 2009
>
> ### **W2-3:** Occlusions.
> Our evaluation explicitly covers diverse occlusion modes, including both **natural and simulated** cases.
> * Real-world benchmarks inherently contain natural occlusions from scene geometry and camera viewpoints
> * We further add a simulation-based occlusion experiment **(Table 3, Lines 432-445)**, which was **not** included in prior baselines.
>
> ### **W2-4:** Intra-class variation.
> Existing benchmarks **(Table 1–3)** share overlapping categories (e.g., Laptop, Scissors, Drawer) with substantial geometric variation across datasets. Evaluating across them thus provides implicit cross-dataset validation of category-level generalization.
> Our method achieves consistent performance across these diverse instances and articulation patterns, confirming robustness to intra-class variation rather than dataset-specific bias.
>
> ## **W3: Ambiguity of joint identification under symmetric or low-texture conditions.**
>
> We sincerely thank the reviewer for raising this critical point.
> **First**, we acknowledge that structural ambiguities, such as closed configurations and symmetry, are challenging to address. But we also observe that color images can provide semantic information, enabling a better understanding of symmetry and closed configurations than geometry-only approaches **(as discussed in Lines 47-50)**. **Second**, trained on extensive and diverse datasets, the VGGT model provides strong geometric modeling even on low-texture surfaces. **Finally**, this motivates us to develop a semantic-geometric fusion module that leverages both strong semantic cues from the image and geometric cues from the VGGT model.
>
> Although resolving the structural ambiguities remains a challenging problem, we have validated the effectiveness of this fusion strategy and achieved the SOTA performance with consistent improvements over the previous geometry-only models **(Table 1–3)**.

---

> ### Author Response · Authors · 2025-11-22
>
> We would like to once again express our sincere appreciation to Reviewer m48d for the thoughtful and detailed feedback. Detailed responses to the main concerns are provided below.
>
> ## **Q1: Determination of joint type.**
>
> As clarified in Sec. 3.1 Task Formulation **(Lines 147–157)**, we follow the **standard protocol**, where part count and joint types are predefined per category.
> The task therefore focuses on estimating continuous articulation parameters and transformations. This canonical setup ensures fair and consistent benchmarking across all datasets.
>
> ## **Q2: Mean-shape bias and axis skew from learnable template.**
>
> We respectfully thank the reviewer for raising this point with such technical depth.
> We agree that the template may introduce mean-shape bias. In fact, **to balance this**, we preserve the canonical frame as a shared prior, while explicitly avoiding template-induced averaging through **two mechanisms** not explored in prior work:
> * **semantic–geometric fusion** injects semantic cues into geometric features, preserving instance variation and preventing template averaging;
> * **physically-consistent augmentation (Sec. D.2, Lines 909–929)** jointly scales predicted and canonicalized shapes, preventing scale-induced artifacts and template bias toward a fixed global scale.
>
> Together, these designs prevent the category template from dominating reconstruction
> and enable scale-invariant, instance-aware learning, effectively avoiding variance collapse.
>
>
>
> To quantify potential mean-shape averaging, we compute a variance ratio $\rho_{\text{var}}$ between predicted reconstructions $\{\hat{S}_i\}$ and ground-truth canonical shapes $\{S_i\}$. We use IoU as the shape similarity metric, and define $\bar S$ as the mean canonical shape that minimizes the average IoU distance to all shapes in the set.
>
> The variance ratio is defined as
> $\rho_{\text{var}} = \frac{Var_{IoU}(\hat{S_i})}{Var_{IoU}(S_i)}$.
> The IoU-based variance is computed as
> $Var_{IoU}(S_i) = \frac{1}{N}\sum_i d_{IoU}(S_i, \bar S)$.
> A smaller $\rho_{\text{var}}$ indicates reduced instance diversity and potential shape collapse.
>
> As shown in **Rebuttal Table B**, $\rho_{\text{var}}$ remains within $0.85$–$1.10$ across all categories,
> confirming that our framework can still preserve instance-level diversity and does not collapse to the template.
>
>
>
> **Rebuttal Table B. Variance ratio ($\rho_{\text{var}}$) across datasets.**
>
>
> | **Datasets** | **Laptop (OP-Align)** | **Suitcase (OP-Align)** | **Safe (HOI4D)** | **Scissors (HOI4D)** | **Eyeglasses (PartNet-Mobility)** | **Drawer (PartNet-Mobility)** |
> |:-------------|:-----------:|:-------------:|:---------:|:--------------:|:---------------:|:-----------:|
> | **$\rho_{\text{var}}$** | 0.98 ± 0.04 | 1.06 ± 0.05 | 0.94 ± 0.03 | 0.87 ± 0.06 | 1.03 ± 0.04 | 0.96 ± 0.05 |
>
>
>
> ## **Q3 & W1: Attribution of gains between backbone prior and the proposed framework.**
>
>
> We include additional ablation studies regarding the proposed recursive prompting and semantic-geometry pyramid fusion, with the results shown in **Rebuttal Table C**. It shows that **directly using the pre-trained VGGT backbone results in worse performance than the current SOTA** (OP-Align), whereas incorporating our recursive prompting improves results. Our full model with recursive prompting and the Cross Semantic–Geometry Pyramid achieved the best results, outperforming the SOTA. This indicates that although VGGT provides strong monocular 3D priors, its rigid-object pretraining leaves it unaware of articulation **(Lines 56–59)**. This synergy between recursive prompting and the Cross Semantic–Geometry Pyramid is essential for transforming VGGT’s rigid prior into an articulation-aware representation.
>
>
> **Rebuttal Table C. Consistent gains from our framework over geometry-only models.**
>
> | ID  | **Method**                        | **Joint Precision (mAP↑)** | **Part Precision (mAP↑)** |
> |:---:|:----------------------------------|:---------------------------:|:---------------------------:|
> | (A) | **Geom. Only (VGGT)**                 | 14.19, 75.29, 87.23         | 15.56, 53.94, 70.02         |
> | (B) | Geom. Only (w/ Recursive Prompting, see Table 6)       | 14.62, 77.17, 89.23         | 16.28, 55.46, 72.05         |
> | (C) | **OP-Align** (see Table 1)                | 15.54, 79.33, 92.36         | 18.33, 57.17, 73.90         |
> | (D) | Full Model (see Table 6)              | 17.72, 80.59, 93.33         | 19.81, 58.96, 75.47         |
>
>
> **[Dear Reviewer m48d]** We sincerely appreciate your careful review and high standards. Your thoughtful feedback has been valuable in improving the overall clarity and quality of our work, and we remain open to any further questions or suggestions you may have.

---

> ### Comment · Reviewer_m48d · 2025-11-26
>
> Thanks to the authors for their detailed response and additional experiments, which have clarified some of my initial concerns. However, two major issues remain unresolved regarding the interpretability of the method and the scalability of the evaluation:
> 1.  I understand that VGGT is pre-trained to extract rigid features. However, it remains unclear why the proposed recursive prompting can effectively transform a rigid prior into an articulation-aware representation. While the experiments show it works, I am concerned that the performance gains might stem primarily from the increased parameter count in the prompt generator rather than the recursive design itself.
> 2. I have reservations regarding the scalability arguments:
> + I am not fully convinced by the argument that "real-world articulations are largely rotational" . Prismatic joints are ubiquitous and present unique challenges, such as severe occlusion. As shown in Table 3, the performance on Drawers is significantly lower than on Laptops, suggesting the method struggles with these harder cases.
> + The category coverage across the evaluated benchmarks seems somewhat restricted. Assuming fairness can be maintained, I believe broadening the evaluation scope to include more diverse categories would be valuable to demonstrate the method's true scalability.

---

> > ### Author Response · Authors · 2025-11-30
> >
> > We thank the reviewer for the insightful question. The concerns are addressed as follows.
> > ## **Q″1: Why recursion works beyond parameter count.**
> >
> > We note that although VGGT is trained on rigid objects and scenes, it has learned generic geometric features that are also critical for understanding articulation. Our **key insight** is that **rigidity and articulation-awareness are not separate representations but two ends of a shared geometric feature manifold**.
> >
> > Recursive prompting provides a principled way to traverse this manifold through dual-gated feedback (**Lines 216–229**), progressively aligning the rigid subspace with articulation-induced deformations (**Lines 235–237**).
> > We elaborate on this from two complementary perspectives: **(1) mechanistic interpretation** and **(2) empirical validation.**
> >
> >
> > ### **(1) Mechanistic interpretation.**
> >
> > **First, structural design (parameter efficiency).**  Recursive prompting introduces no additional parameters; it reuses the same generator in a truncated fixed-point iteration, performing residual refinements within the same latent subspace rather than adding new layers.
> > Dual-end residual gates at the backbone boundaries enable controlled feature modulation while maintaining VGGT stability.
> >
> > **Second, mechanistic effect (why recursion works).**  The residual prompts operate along dual gradient paths:
> > $\tilde{\eta} = (1-\beta_1)\eta + \beta_1 P_{\text{in}}(\eta), \tilde{z} = (1-\beta_2)z + \beta_2 P_{\text{out}}(z)$.
> >
> > During backpropagation, gradients flow through both the frozen backbone and the gated prompt branches, where $\beta_1$ and $\beta_2$ dynamically modulate spatial and semantic cues.  This dual-path feedback progressively aligns the rigid feature subspace with articulation-induced deformations, guiding recursion toward a motion-consistent equilibrium without increasing parameters.
> >
> > **Finally, convergence dynamics (rigid → articulation-aware).**  Part boundaries become sharper and geometric alignment more consistent (**Fig. 4, Lines 483–500**) as recursion proceeds (1 → 12), confirming that recursive prompting incrementally refines geometric structure without extra supervision.
> >
> > This process effectively serves as a depth-equivalent fine-tuning: each residual iteration propagates motion-consistent deformation signals across the feature manifold, guiding rigid features toward an articulation-aware equilibrium.
> >
> > ### **(2) Empirical validation.**
> >
> > Controlled experiments (**Lines 467–482**) verify this mechanism through two analyses (impact of recursion rounds and comparison with layer stacking).
> >
> > * **Fixed-parameter, single-layer setting (Lines 474–481).** With constant model capacity, increasing recursion rounds consistently improves segmentation performance (**Fig. 3**).
> > * **Layer stacking (Lines 467–472, Table 5).** Even stacking up to four layers yields lower performance than deeper recursion under the same parameter count.
> >
> > These results confirm that the observed gains originate from the recursive mechanism rather than parameter count, validating recursion as a depth-equivalent process that enhances expressivity without additional parameters.

---

> ### Author Response · Authors · 2025-11-30
>
> ## **Q″2-1: Concern about limited prismatic joints and lower performance on Drawer category.**
>
> We thank the reviewer for highlighting the importance of prismatic articulations.
>
> We note that the concern arises from the implicit assumption that our statement implies a priority of rotational joints, whereas our intent was to clarify a **benchmark-level distributional bias**, not an ontological importance.
> Specifically, existing category-level articulation benchmarks contain only a small fraction of prismatic objects (HOI4D 1/9, OP-Align 1/5, PartNet-Mobility 9/46), which **inevitably limits their representation in evaluation**.
>
> Our goal is **not to rebalance datasets**, but to **develop the first self-supervised RGB-only framework** capable of **handling both revolute and prismatic articulations under this realistic distributional constraint.**
>
>
> We fully agree that prismatic cases such as drawers are inherently challenging, involving severe self-occlusion and limited motion cues.
> However, we would like to emphasize that the observed performance gap mainly arises from **intrinsic differences between categories (e.g., drawer vs. laptop)** and **articulation types (prismatic vs. revolute)**, which entail distinct motion patterns and occlusion characteristics, rather than from any limitation of our approach.
> A fair comparison should thus be **made within each articulation category**, where our framework achieves SOTA results across all prismatic and rotational cases.
>
>
> Finally, we appreciate this insightful point highlighting an under-represented aspect of current benchmark design.
> We have **added a discussion** on these dataset biases and articulation-type differences in the revised paper to clarify their evaluation implications (Lines 363–368).
>
> ## **Q″2-2: Expanding the evaluation scope.**
>
> We appreciate the suggestion and agree that expanding the evaluation to more categories is valuable for assessing scalability.
> Our focus, however, is on **how well the framework generalizes across diverse categories rather than the absolute number evaluated**.
>
> **Our evaluation already covers the largest** and most representative articulation benchmarks available to date, and the pool of public datasets remains limited, making further comparisons difficult within the short rebuttal period.
>
> Within these constraints, we still **expanded the experiments** (Lines 429–440), and our current evaluation now includes additional categories spanning a broader range of shapes **(+3 over OP-Align and +6 over EAP)**.
>
> **[Dear Reviewer m48d]** We sincerely hope that our clarifications have addressed your concerns and provided a clear explanation.

---

### Official Review · Reviewer_1e2x · 2025-10-30

**Soundness:** 2
**Presentation:** 2
**Contribution:** 2
**Rating:** 2
**Confidence:** 5

**Summary:**

This paper introduces RECAP, a self-supervised method for single image category-level articulated object pose estimation. To tackle the depth uncertainty problem, the authors exploit a geometry foundation model to learn the corresponding complete point cloud for the input object, with the proposed recursive prompt for adapting articulated objects. Then an alignment method is used for optimizing the per-part 6D pose using the reconstructed point cloud into the RGB image.

**Strengths:**

1. The authors solve the pose estimation problem using an only RGB image, which is promising.
2. The technique presentation is sound and convincing.
3. Enough experiments are provided.

**Weaknesses:**

1.To address the depth missing problem, the authors employ a geometric foundation model for point cloud learning. However, the comparison of point cloud reconstruction with methods that do not utilize such foundation models is arguably unfair. Although the RECAP method introduces external knowledge for the pose estimation task, it only achieves marginal improvements.
2.Several relevant works on category-level articulation pose estimation are not adequately cited or discussed, such as R2-Art (AAAI 2025), U-COPE (ECCV 2024), and "Toward real-world category-level articulation pose estimation" (TIP 2022). Additionally, the well-established render-and-compare methodology, widely used for single-image pose estimation, is also overlooked in the discussion.
3.The authors utilize the OP-Align dataset as a benchmark; however, its scale is relatively limited, encompassing only four categories. Given that existing datasets contain over 2,000 objects across numerous categories, the selection of merely four categories appears insufficient.
4.The evaluation does not include two widely recognized articulation datasets—ArtImage and ReArtMix. The authors are encouraged to provide an explanation for this omission.

**Questions:**

please refer to weaknesses.

---

> ### Author Response · Authors · 2025-11-22
>
> We sincerely thank Reviewer 1e2x for the constructive and valuable comments. The concerns are addressed as follows.
> ## **Q1: Fairness of foundation model comparison and marginal improvement.**
>
>
> **(1) Comparison fairness**. We respectfully clarify that our comparison is fair and, in fact, emphasizes a more challenging yet practically relevant setting. While prior methods rely on explicit depth input, our framework instead explores how a geometric foundation model can be effectively utilized to perform articulated reasoning under an **RGB-only constraint**, without any depth input or supervision. Importantly, as discussed in **Lines 56–59** and supported by **Table 6(A)**, **directly using the pre-trained VGGT model for articulation estimation underperforms the SOTA method**, indicating that the observed improvement stems from our proposed recursive prompting and semantic–geometry fusion rather than the foundation model itself.
>
>
>
> **(2) Marginal improvement.** Please refer to **Rebuttal Table A**, which summarizes the results. As discussed in **Lines 56–59** and quantified in **Table 6(A)**, the base VGGT, trained on rigid scenes, struggles to model articulated motion, resulting in suboptimal performance.
> Our framework builds articulation reasoning on top of these rigid priors, achieving an average gain of +7\%  across three benchmarks.
> What appears marginal numerically represents a substantial qualitative leap, enabling articulated pose estimation from single RGB images for the first time.
> This consistent improvement confirms that the advantage stems from our framework’s adaptation of rigid 3D priors to articulated reasoning, rather than from the backbone itself.
>
>
> **Rebuttal Table A**. Consistent gains from our framework over geometry-only models.
> | ID  | **Method**                        | **Joint Precision (mAP↑)** | **Part Precision (mAP↑)** |
> |:---:|:----------------------------------|:---------------------------:|:---------------------------:|
> | (A) | **Geom. Only** (VGGT)                 | 14.19, 75.29, 87.23         | 15.56, 53.94, 70.02         |
> | (B) | Geom. Only (w/ Recursive Prompting, see Table 6)       | 14.62, 77.17, 89.23         | 16.28, 55.46, 72.05         |
> | (C) | **OP-Align** (see Table 1)                | 15.54, 79.33, 92.36         | 18.33, 57.17, 73.90         |
> | (D) | Full Model (see Table 6)              | 17.72, 80.59, 93.33         | 19.81, 58.96, 75.47         |
>
>
> ## **Q2: Missing related works and overlooked render-and-compare paradigm.**
>
> We thank the reviewer for highlighting these works.
> While we acknowledge their relevance, these works (R2-Art, U-COPE, TIP 2022) follow fully supervised settings relying on depth or 6D pose labels, which fundamentally differ from our self-supervised setup.
> Our framework instead learns articulation reasoning directly from RGB images without geometric supervision, addressing the label-dependency that limits those methods.
> We have added a discussion in the Related Work **(Lines 91-97)** and will include direct comparisons once the official implementations become available.
>
> ## **Q3: Limited benchmark coverage.**
>
> Our evaluation spans three standard datasets, covering the **largest articulation estimation benchmark to date**, compared to the most recent baselines such as OP-Align and EAP. In total, it includes more than **48,000 objects** (over 6,000 in OP-Align and 41,000 in PartNet-Mobility), and we further extend the articulation taxonomy by one additional category compared to OP-Align and four additional categories compared to EAP.
>
> Moreover, we further evaluate occlusion robustness through **simulation-based tests** **(Table 3, Lines 432–445)**, a factor **not** covered in prior baselines.
>
> Together, these settings extend beyond existing benchmarks in both diversity and difficulty, providing a more comprehensive testbed for articulation estimation.
>
> ## **Q4: Missing evaluation on ArtImage and ReArtMix.**
>
> We appreciate the reviewer’s observation and clarify this aspect below.
>
> * ArtImage dataset: ArtImage dataset is **already included** in our evaluation, as it is derived from PartNet-Mobility **(Sec. F, Lines 970–971)**.
> We follow the same processing protocol as ArtImage, which ensures that our experiments on PartNet-Mobility cover the same object categories and joint definitions.
> * ReArtMix: ReArtMix has not yet been officially released. However, as you suggested, we have added an important discussion of this dataset in the revised Related Work and will include corresponding evaluations once it becomes publicly available.
>
> **[Dear Reviewer 1e2x]** We sincerely appreciate the reviewer’s attention to detail and high standards. We value your feedback and look forward to addressing any further questions or concerns.

---

> > ### Comment · Reviewer_1e2x · 2025-11-27
> > **raise my score**
> >
> > The response has addressed most of my concers so I raise my score to 6.

---

> ### Author Response · Authors · 2025-11-28
>
> We thank the reviewer for the response. We are very pleased to see that all concerns have been addressed, and **we greatly appreciate the decision to raise the score to 6, which was made before the leak incident occurred**.

---

### Official Review · Reviewer_DKcg · 2025-10-31

**Soundness:** 3
**Presentation:** 4
**Contribution:** 4
**Rating:** 8
**Confidence:** 2

**Summary:**

This paper presents ReCAP, a self-supervised framework for category-level articulated object pose estimation from a single RGB image — a task that traditionally requires depth or multi-view supervision. ReCAP adapts a large geometry foundation model (VGGT) using a novel Recursive Residual Prompting mechanism, which refines prompts through iterative recursion and stabilizes them via residual injection, introducing less than 1% additional parameters. This enables parameter-efficient adaptation of rigid-object priors to articulated settings.

To address occlusion and symmetry ambiguities, the paper further introduces a Cross Semantic–Geometry Pyramid (X-SGP) module that hierarchically fuses semantic and geometric cues. Experiments on OP-Align, HOI4D, and PartNet-Mobility show that ReCAP achieves state-of-the-art performance among self-supervised methods and even surpasses some supervised RGB-D baselines.

**Strengths:**

* The proposed Recursive Residual Prompting (RRP) is a well-motivated and technically elegant solution to two core issues in applying prompt tuning to large geometry backbones such as VGGT: (1) limited capacity of shallow prompts to capture complex articulation patterns, and (2) instability caused by VGGT’s dynamic token reconfiguration.
* The proposed Cross Semantic–Geometry Pyramid (X-SGP) effectively fuses semantic and geometric cues via adaptive FiLM modulation and multi-scale refinement. Ablation studies show clear performance drops when removing pyramid layers or FiLM components, confirming their necessity in handling occlusion, symmetry, and fine-grained part alignment.
* The authors provide quantitative ablations for recursion depth, prompt placement (input/output), and parameter scaling, showing that the recursive approach achieves comparable or better performance than multi-layer stacking with only 0.8% additional parameters. This level of analysis supports the soundness and reproducibility of the proposed design.

**Weaknesses:**

* The work thoughtfully adapts prompt tuning and DEQ-style recursion to articulated pose estimation, which is a valuable and nontrivial contribution. Still, the methodological core builds on established ideas, with innovation mainly in integration and application rather than new theoretical development.
* Despite its parameter efficiency (<1% trainable), the recursive refinement introduces noticeable latency (≈13 FPS vs. 41 FPS in baselines). Discussion of this trade-off or adaptive recursion strategies would improve clarity on practical feasibility.

**Questions:**

* The paper could better illustrate how recursive prompting reshapes geometric or semantic representations. For example, visualizing token attention or feature evolution across recursion steps would clarify how the mechanism contributes to improved articulation reasoning beyond empirical gains.

---

> ### Author Response · Authors · 2025-11-22
>
> We sincerely thank Reviewer DKcg for the constructive feedback and for the deep technical insight reflected in the review. All comments have been carefully addressed in detail as follows.
>
> ## **W1: Limited theoretical novelty.**
>
> We appreciate the reviewer’s thoughtful assessment.
> We agree that broader theoretical development is important, and we want to point out that our work is theoretically motivated and establishes a new task paradigm rather than an integration.
> Specifically, we introduce, to the best of our knowledge, the **first self-supervised articulation framework trained on single RGB images**.
> The theoretically driven recursive prompting unlocks the foundation model’s latent 3D priors for articulation reasoning. This recursive prompting strategy can be applied in general to other Transformer-based foundation models for efficient fine-tuning.
> Finally, the proposed method shifts articulation estimation from conventional **geometry-centric modeling pipelines to single-image, prior-driven reasoning.**
>
> ## **W2: Latency–efficiency trade-off in recursive refinement.**
>
>
> As reported in **Table 3 (Lines 432–445)**, our method adds only 52 ms per frame over OP-Align, while EAP even runs slower.
> This slight latency is the natural cost of achieving single-image deployment with higher accuracy, a capability that prior depth-based methods cannot achieve. Moreover, the number of recursive refinement steps is user-controllable **(Lines 209–215; Lines 474-482, Fig. 3)**, making it a flexible design choice.
>
> ## **Q1:  Visualization of evolution in recursive prompting.**
>
> We sincerely thank the reviewer for this valuable suggestion. As you suggested, we have added visualization and analysis across recursion steps in Sec. 4.3 **(Lines 483-500, Fig. 4)**, showing that recursive prompting progressively sharpens part boundaries and aligns local geometry into semantically consistent configurations. This visual evolution confirms that recursion acts as a depth-equivalent refinement mechanism, continuously improving feature representations until convergence, yielding more reliable part structure and directly benefiting articulated object pose estimation.
>
> **[Dear Reviewer DKcg]** We sincerely thank you for your thorough and insightful review. Your constructive feedback has been invaluable in improving both the technical clarity and overall presentation of our work.

---

### Public Comment · ~Hang_Zhang25 · 2025-11-17
**What the difference is between Cross Semantic–Geometry Pyramid and DPT-Head used in VGGT?**

Also, VGGT does have the camera head, right?

---

### Author Response · Authors · 2025-11-22
**General Response**

Thank you for providing constructive insights on our paper and for taking the time to review the paper. We have revised the manuscript as outlined below to incorporate reviewers' comments, with the revised manuscript written in **blue**.

1. We have added visualization of recursive prompt refinement across recursion rounds in Sec. 4.3 Ablation **(Lines 483-500; Fig. 4)**.

2. We have updated additional relevant works in Sec. 2 Related Work **(Lines 91-97)**.

3. We have added a discussion clarifying that the imbalance between rotational and prismatic articulations reflects benchmark bias and offers insights for future benchmark design **(Lines 363–368)**.

4. We have expanded the experiments **(Table 3; Lines 429–440)** to include two additional categories.

We greatly appreciate the reviewer’s valuable suggestions. We have highlighted relevant rebuttal sections, and figures in blue.
Additional changes will be incorporated in a subsequent revision.

---

> ### Author Response · Authors · 2025-12-02
> **General Response (Updated)**
>
> Dear Reviewers,
>
> We sincerely thank you for the valuable feedback and constructive suggestions. Following your comments, we have incorporated the **latest updates** into the main paper (**Nov 30** revision), with all revisions highlighted in **blue**.
>
> Best regards,
>
> The Authors

---

### Author Response · Authors · 2025-11-30
**Rebuttal Summary for the AC**

Dear AC,

We sincerely appreciate your effort in coordinating the review process. Please note that **before the information leak occurred**, we had already clarified the concern of Reviewer 1e2x during the rebuttal, which raised the overall **score from (8, 2, 4) to (8, 6, 4)**. To ease your review, we include a brief rebuttal summary below.

1. Reviewer **DKcg (Score: 8, “Good paper”)**: We have added visualization of recursive prompting as suggested.
2. Reviewer **1e2x (Initial: 2 → Updated: 6, comment updated before the leak incident)**: The main concerns were the performance gain and benchmark coverage, both of which have been clarified in the rebuttal.
3. Reviewer **m48d (Score: 4; 12 questions in total)**: We addressed 9 out of 12 concerns in the first-round rebuttal, and are now fully clarifying the **remaining 3 in the second round**:
* **Q''1 Explanation of rigidity and articulation-awareness**: we elaborate on this from two complementary perspectives: (1) mechanistic interpretation and (2) empirical validation.
* **Q''2-1 Concern about limited prismatic joints**: We clarify that **this reflects a benchmark-level bias**; our goal is **not to rebalance datasets**, but to build a self-supervised RGB-only framework that handles both revolute and prismatic articulations.
* **Q''2-2 Scalability evaluation**: **While our evaluation already covers the largest articulation benchmarks** used in recent studies, we further **expanded the experiments** in response to the reviewer’s suggestion. Given the limited datasets and time, we extended the evaluation wherever possible.

Before the leak was reported, the overall score had increased from (8, 2, 4) to **(8, 6, 4), avg. 6, with further increases likely**. We would be grateful if AC would consider our contributions:
* A 0.8%-parameter, depth-equivalent recursive prompting mechanism that unlocks VGGT’s articulation reasoning;
* The first self-supervised articulated object pose estimation framework from a single image.

We hope this paradigm shift will inspire the community.



Sincerely,

ReCAP Authors

---

### Meta-Review · Area_Chair_kCqC · 2026-01-05

**Summary:**

One reviewer's concerns:
The interpretability of the proposed recursive prompting remains unclear, as it is not convincingly shown why it transforms rigid features into articulation-aware representations rather than simply benefiting from increased parameters. Second, the scalability claims are insufficiently supported, with limited category coverage and weaker performance on challenging prismatic articulations, raising doubts about generalization to diverse real-world settings.

**Reviewer Concerns:**

The interpretability of the proposed recursive prompting remains unclear, as it is not convincingly shown why it transforms rigid features into articulation-aware representations rather than simply benefiting from increased parameters. Second, the scalability claims are insufficiently supported, with limited category coverage and weaker performance on challenging prismatic articulations, raising doubts about generalization to diverse real-world settings.

**Reviewer Scores:**

The first reviewer felt that the paper lacks theoretical novelty and latency introduced by recursive refinement though the reviewer gave a score of 8, he is a low confiden. The two other reviewes gave scores of reject (2). and border reject (4). One reviewer suddenly moved to 6 from 2 after rebuttal. Other reviewers did not change the scores.

---

### Decision · Program_Chairs · 2026-01-26

Reject